# Stigma of Opioid Use Disorder and Its Indirect Effects on Student Pharmacists’ Perceptions and Attitudes

**DOI:** 10.3390/pharmacy8030144

**Published:** 2020-08-14

**Authors:** Elizabeth A. Hall, Alina Cernasev, Umida Nasritdinova, Michael P. Veve, Kenneth C. Hohmeier

**Affiliations:** Department of Clinical Pharmacy and Translational Science, College of Pharmacy, The University of Tennessee Health Science Center, Nashville, TN 37211, USA; liz.hall@uthsc.edu (E.A.H.); unasritd@uthsc.edu (U.N.); mveve1@uthsc.edu (M.P.V.); khohmeie@uthsc.edu (K.C.H.)

**Keywords:** opioid use disorder, opioid stigma, addiction, pharmacy student

## Abstract

Objectives: Pharmacists play a vital role in serving patients during the ongoing nationwide opioid epidemic, and so it is also critical to educate the next generation of pharmacists on opioids and opioid use disorder (OUD). The primary objective of this study was to quantitatively characterize student perceptions of opioid use and the stigma associated with OUD. Secondary aims were to determine whether differences in perceptions exist based upon the student’s year in the Doctor of Pharmacy program or employment in a community pharmacy. Methods: First-, second-, third-, and fourth-year student pharmacists voluntarily completed an electronic survey regarding perceptions of opioid use and stigma associated with OUD. Results: Of the 9 survey items, students were most uncomfortable referring patients to community resources for addiction support and/or treatment (25.3% comfortable or very comfortable). Students working in a community pharmacy were significantly more comfortable talking to patients attempting to refill opioids early and providing opioid counseling as compared to their peers not working in community pharmacy. Fourth-year students reported a higher level of comfort talking to a patient attempting to refill an opioid prescription early, counseling a patient on an opioid prescription, and providing information about alternatives to opioids. Third-year students responded most favorably to the items regarding how well the curriculum has prepared them to interact with patients taking opioids and those with OUD. Conclusions: These findings reveal that students are comfortable counseling on opioids and discussing alternative options. Differences in perceptions were observed based upon the student’s year in the program and whether or not they were employed in a community pharmacy setting.

## 1. Introduction

The United States (US) opioid epidemic has been a paramount public health issue due to its significant societal consequences, including increased mortality, neonatal abstinence syndrome, as well as sharp increase in treatment for opioid use disorder (OUD). Overall, its cost to society amounts to over USD 78.5 billion per year [1]. From 1999 to 2011, the distribution rate of oxycodone in the US quadrupled and hydrocodone distribution tripled. During the same time, opioid-related deaths increased by 400%, claiming over 750,000 lives [2,3]. In 2018 alone, opioid-related deaths totaled 46,802, accounting for 70% of all drug overdose-related deaths [4].

Despite evidence showing its chronicity, the stigmatization of OUD continues to pervade all aspects of society today. Stigma, as defined by Link and Phelan, is the co-occurrence of labeling, stereotyping, separation, status loss, and discrimination, and it is one of the leading predictors of poor health outcomes in those who suffer from OUD [5,6]. Stigma presents itself in many forms, including enacted, structural, and internalized stigma. Though the list is extensive, few instances include: utilization of negatively-connotated language such as “junkie” or “clean” versus “dirty” when referring to those with OUD, placing numerous regulatory and institutional barriers to access medication-assisted therapy (MAT), and historical labeling of these patients as “difficult to treat” [7,8,9].

Stigmatizing policies and language have heavily influenced how the public perceives OUD. A large-scale survey of public perception of opioid misuse found that higher stigma ratings were associated with greater support for punitive policies rather than proactive public health initiatives, such as Medicaid expansion to cover MAT treatment [10]. Moreover, amongst surveyed medical providers, 66.4% believed those who are addicted to pain medications are more dangerous than the general population and 76.6% were unwilling to work closely with someone who is affected by OUD [11]. As a result of mass distribution of misconceptions and barriers to therapy, those who consume opioids struggle with internalized stigma, or the attachment of public perceptions to oneself, which has been associated with increased non-fatal overdoses, continued substance abuse, and deteriorating mental health [12,13,14].

Pharmacists have been actively mitigating the effects of the epidemic through screening (e.g., drug utilization reviews, prescription drug monitoring database reviews) for opioid diversion as well as furnishing naloxone to those who are at high risk of opioid-related adverse effects [15]. However, both real and patient-perceived stigma related to opioid use has been previously reported in the literature and may present further barriers to treatment [16,17]. It is crucial to identify aspects that need additional strengthening and educate the next generation of pharmacists. Currently, studies regarding student pharmacists’ perceptions of opioids, OUD, and the stigma associated with OUD are limited. Thus, the aim of this study was to quantitatively characterize student perceptions of opioid use and stigma associated with OUD. Secondary aims were to determine whether differences in perceptions exist depending upon the student’s year in the Doctor of Pharmacy program and if there are differences in perceptions for students employed in a community pharmacy setting versus those who are not.

## 2. Methods

A survey was selected because of its convenience, cost-efficiency, anonymity, time-saving ability, and quick results [18]. It was constructed using concepts from the Stigma Conceptualization framework proposed by Link and Phelan [5]. Additional stigma-related questions were developed using the related literature [19,20]. To better understand the concept of stigma, participants were presented with brief statements describing situations in which they could portray stigmatizing behavior towards a colleague who suffers from addiction [20]. Participants were asked to report how likely they would avoid a colleague with addiction, as well as their comfort level in continuing a friendship with a colleague diagnosed with addiction. The survey also asked if the student interacted with patients who had multiple opioid prescriptions or patients who sought early refills and the student’s level of comfort counseling patients on opioid prescriptions. Example questions include: “How comfortable do you feel talking to patients who request to fill opioid prescriptions earlier than 24 h before they are due?” and “How well did the UTHSC curriculum prepare you to dispense and counsel on opioids?”

This study consisted of students voluntarily completing a survey regarding perceptions of opioid use and stigma associated with OUD. Students at the University of Tennessee Health Science Center (UTHSC) College of Pharmacy who were enrolled in a Doctor of Pharmacy program (*n* = 704) were eligible to participate. The survey was administered from December 2019 until March 2020. Survey respondents included students in their first (P1), second (P2), third (P3), and fourth (P4) professional year. The 18-item survey was delivered electronically, and all responses were anonymous. The survey responses were also captured and stored electronically. The UTHSC Institutional Review Board granted exemption approval for this study (19-06977-XM).

The survey topics included students’ beliefs and attitudes about opioid prescriptions and counseling, the stigma associated with OUD, patients at higher risk of OUD, and whether patients would benefit from referral for MAT services. There were nine survey items assessing these topics, and all items were closed-ended questions. In the survey, participants were asked to indicate their level of comfort for four items regarding opioid use and one item regarding stigma using a 5-point Likert Scale (1 = very comfortable to 5 = very uncomfortable) [18]. Two additional items used a 3-point Likert Scale (“very well,” “well,” “not well,” and “unsure”) to assess perceptions of how well the curriculum at the College of Pharmacy prepared students to dispense and counsel on opioids as well as screen and interact with patients at risk or with OUD. Another two survey items assessed stigma manifestation, specifically student likelihood to avoid classmates with OUD and likelihood to counsel patients with suspected OUD on the importance of being “addiction-free,” using a 5-point Likert scale (1 = very likely to 5 = very unlikely) [5,18]. At the end of the survey, participants answered nine questions regarding demographics and pharmacy work history. Work history questions included the setting (i.e., community, hospital, or other pharmacy setting) and duration of employment. Survey participants who responded as working in a community pharmacy setting were also asked how often they interact with patients receiving opioids and how often patients ask if the pharmacy carries certain opioids. The survey participant could choose not to respond to any questions within the survey.

Data were analyzed using SPSS for Windows, version 25.0 (IBM Corporation, Armonk, NY, USA). Descriptive statistics were calculated for all variables (i.e., median and range for nonparametric numeric data and frequencies and percentages for all nominal and ordinal data). Between group differences in demographics were determined using chi-square tests for all nominal data and Mann–Whitney or Kruskal–Wallis tests for all nonparametric numeric data (two-group comparison or four-group comparison, respectively). To determine differences in perceptions between the four student classes, Kruskal–Wallis tests were used. To compare differences in perceptions for those students who were working in a community pharmacy setting versus those who were not, Mann–Whitney tests were used. All tests were two-tailed, and an a priori alpha level of 0.05 was used to determine significance.

## 3. Results

A total of 244 (34.7%) eligible participants responded to the survey. Respondent demographics are summarized in Table 1.

Figure 1 summarizes student responses to the nine survey items regarding perceptions of opioid use and stigma associated with OUD.

Survey responses were compared between students currently working in a community pharmacy setting (*n* = 150) versus those not currently working in a community pharmacy setting (*n* = 94). Between groups comparisons of demographic data are presented in Table 2. Students currently working in a community pharmacy setting indicated being more comfortable talking with patients attempting to refill opioids more than 24 h early (*p* < 0.001) and counseling a patient on an opioid prescription (*p* = 0.001) as compared to their peers who do not work in a community pharmacy setting. There was no significant difference between the two groups of students in responses regarding comfort with providing information regarding alternatives to opioids (*p* = 0.463) or referring a patient to community resources for addiction support or treatment (*p* = 0.370). Students who work in a community pharmacy setting felt that the UTHSC College of Pharmacy curriculum prepared them to dispense and counsel on opioids (*p* = 0.024) as compared to their peers who do not work in community pharmacy; however, there was no significant difference in how well the curriculum prepares students to screen and interact with patients who have or are at risk of OUD (*p* = 0.081). Likewise, there was no significant difference in responses regarding likelihood to avoid a classmate if the student found out he/she was abusing opioids (*p* = 0.910). Students not currently working in a community pharmacy setting noted a higher likelihood of explaining the importance of “addiction-free” to a patient who usually picks up multiple opioid prescriptions (*p* = 0.003).

Survey responses were compared for P1 students (*n* = 33), P2 students (*n* = 82), P3 students (*n* = 78), and P4 students (*n* = 51). Between groups comparisons of demographic data are presented in Table 3. There was a significant difference in survey responses between the different classes for the following items: comfort talking to patients who are attempting to refill opioid prescriptions more than 24 h early (Kruskal–Wallis mean rank score of 117.45 for P1s, 138.65 for P2s, 121.46 for P3s, and 94.08 for P4s; *p* = 0.003), comfort counseling a patient on an opioid prescription (Kruskal–Wallis mean rank score of 142.40 for P1s, 141.60 for P2s, 106.92 for P3s, and 96.13 for P4s; *p* < 0.001), comfort providing information about alternatives to opioids (Kruskal–Wallis mean rank score of 139.76 for P1s, 138.33 for P2s, 108.95 for P3s, and 99.93 for P4s; *p* = 0.001), as well as how well the curriculum prepared the student to dispense and counsel on opioids (Kruskal–Wallis mean rank score of 122.19 for P1s, 153.23 for P2s, 100.29 for P3s, and 104.87 for P4s; *p* < 0.001) and to interact with patients at risk of or with OUD (Kruskal–Wallis mean rank score of 129.42 for P1s, 151.52 for P2s, 97.58 for P3s, and 107.23 for P4s; *p* < 0.001). There was no significant difference in survey responses between the different classes regarding comfort referring a patient to community resources for addiction support or treatment (*p* = 0.945), comfort level in continuing a friendship with a classmate who is abusing opioids (*p* = 0.582), likelihood to avoid a classmate who is abusing opioids (*p* = 0.620), or likelihood to explain the importance of “addiction-free” to a patient who usually picks up multiple opioid prescriptions (*p* = 0.335). 

## 4. Discussion

This study is among the first to focus on student pharmacists’ attitudes and behaviors towards stigma associated with OUD, particularly when interacting with and counseling patients who are prescribed opioids. The results described within this suggest that student pharmacists feel comfortable counseling patients on opioids and discussing alternative options. Many students report being frequently asked by patients if the pharmacy carries a particular opioid, which indicates that many student pharmacists are exposed to patients who are prescribed opioid medications [21,22]. These results must be interpreted while also considering that the US has seen an alarming increase in the number of opioids prescribed by the healthcare professionals and dispensed by pharmacists in the last decade. Specifically, Tennessee has experienced the consequences of mass opioid distribution more extensively than other states. For example, in 2018, Tennessee’s opioid prescription rates amounted to 94.4 prescriptions per 100 persons versus the national average of 59. per 100 persons, which made it the third highest prescription-averaging state in the country [23,24,25,26]. Despite a 25% decline in opioid prescriptions per person from 2013 to 2017, opioid-related deaths rose from 77 to 590. In addition, heroin-related deaths rose in the same time frame from 50 to 311 deaths [27].

Among this sample of student pharmacists working as interns in a state heavily affected by opioid overprescribing, it seems that they are not comfortable referring a patient to community resources for addiction support or treatment (Figure 1, Q4). As the last healthcare provider before an opioid is dispensed, pharmacists are positioned to play an important role in referring patients to MAT [28]. Previous studies showed that pharmacists can play an important role in screening and referral [22,28]. Additionally, a 2019 pilot study found that pharmacists made referrals when presented with a toolkit with guidance on referral to community resources [29]. Our survey findings suggest that potential training for student pharmacists should be supported by a strong community engagement and partnerships with pharmacies. Furthermore, mutual efforts from pharmacies, practicing pharmacists, and patients are needed to bring about awareness of existing programs and resources for addiction support or treatment.

Differences in survey responses were observed in this study when comparing students who work in a community pharmacy setting versus those who do not. This survey demonstrated that students working in a community pharmacy setting are more comfortable talking with patients about opioids and early refills of opioids as compared to their peers not working in community. This increased level of comfort may be due to the frequency at which the students working in a community setting interact with patients receiving opioid prescriptions. A total of 148 (98.7%) students working in community reported interacting monthly or weekly with these patients. Similarly, 114 (76%) students reported being asked by patients at least monthly whether the pharmacy carries certain opioid medications (Table 1). The authors believe this draws attention to the need for exposure of students to community pharmacy-based patient care with the specific intent of improving student comfortability navigating conversations about opioid use and misuse with patients. Future studies should investigate the required length of exposure in community to promote comfortability so that USA. colleges of pharmacy can best prepare student pharmacists to provide care to these patient populations.

In this study, P4 students reported a higher level of comfort talking to a patient attempting to refill an opioid prescription early, counseling a patient on an opioid prescription, and providing information about alternatives to opioids. This observation may be due to P4 students being on rotations and thus having more exposure to patients who are taking opioids. Third-year students responded most favorably to the two survey items regarding how well the curriculum has prepared them to interact with patients taking opioids and those with OUD. Students learn pharmacotherapy, medicinal chemistry, and pharmacology of opioids in the fall semester of their third professional year in the integrated therapeutics module, which may explain in part why P3 students responded more favorably to perceptions of the curriculum.

Two of the survey items (Q8 and Q9) asked the participant to predict their future behavior, namely, the likelihood of avoiding a classmate abusing opioids and to provide “addiction-free” counseling to a patient who usually picks up multiple opioids. Interestingly, a significant difference in P1, P2, P3, and P4 student responses for these two items was not observed. A poor ability to predict future behaviors may have contributed to the lack of a significant difference.

The study results herein are essential to pharmacy educators because student pharmacists must be competent in their abilities to counsel a patient about opioids, understand the mechanisms of addiction, and screen for opioid overuse, as these actions ultimately lead to harm reduction. Thakur et al. assessed third-year student-pharmacists’ verbal and nonverbal communication skills during a simulated opioid counseling and found that many students lacked confidence in educating patients about opioid-specific risks such as dependence, addiction, or overdose [30]. This study also reinforced the need for additional education and resources on how to best communicate with patients about this sensitive topic [30]. Pharmacy educators should focus their future efforts on harm reduction strategies that include counseling, how to address patients at risk of OUD, and how to screen patients who are at risk of OUD.

The cohort surveyed in this study does not seem to manifest stigmatizing attitudes towards a classmate or a patient exhibiting opioid-seeking behaviors, as shown by student responses to items 5, 8, and 9 on the survey (Figure 1). Stigma can be a major obstacle to obtaining an effective response to OUD [10,31]. Furthermore, as a result of stigma, OUD patients and treatment-seeking individuals face discriminatory hardships in areas of life such as employment, housing, healthcare, relationships, and communication. The complexity of stigma must be addressed at all levels of society [32], and pharmacy educators are well positioned to teach students how to address the stigma from early stages in the pharmacy curriculum.

## 5. Strengths and Limitations

These study results should be interpreted considering some characteristic survey-based research limitations. Participants in the current study voluntarily responded to the survey. Furthermore, this study had a limited response rate from P1 students, which could be due to outreach. However, our response rate for the entire study mirrored other response rates of student pharmacists from other institutions. In addition, the survey was administered to students enrolled at a single institution, which may limit the generalizability of the data. The characteristics of the survey respondents were well-matched to the general population of students enrolled in Doctor of Pharmacy programs, making the likelihood of sample bias in this study low. The survey only solicited if students were currently working and not their prior work history; thus, some students with past experience in community pharmacy may have been classified within the cohort not working in a community pharmacy setting. Prior experience in a community pharmacy may have influenced perceptions of opioid use and consequently participant survey responses, which is a limitation to this study. The survey method used in this study consisted solely of closed-ended questions, which may limit the exploration of beliefs and attitudes. Student focus groups or other similar methods should be considered to further characterize student perceptions of opioid use and the stigma associated with OUD.

## 6. Conclusions

Overall, this study suggests that students generally feel comfortable in counseling patients and discussing alternative options to opioids. This level of comfort tends to increase as the student progresses within the Doctor of Pharmacy curriculum, as shown by the differences in perceptions observed based upon the student’s year in the program. Student pharmacists employed in a community pharmacy setting are more comfortable talking with patients who attempt to refill opioids early and counseling patients on an opioid prescription as compared to their peers who do not work in community pharmacy. Because students reported a relatively low level of comfort referring a patient to community resources for addiction support or treatment, this is an area in which future pharmacy educational efforts should be focused.

## Figures and Tables

**Figure 1 pharmacy-08-00144-f001:**
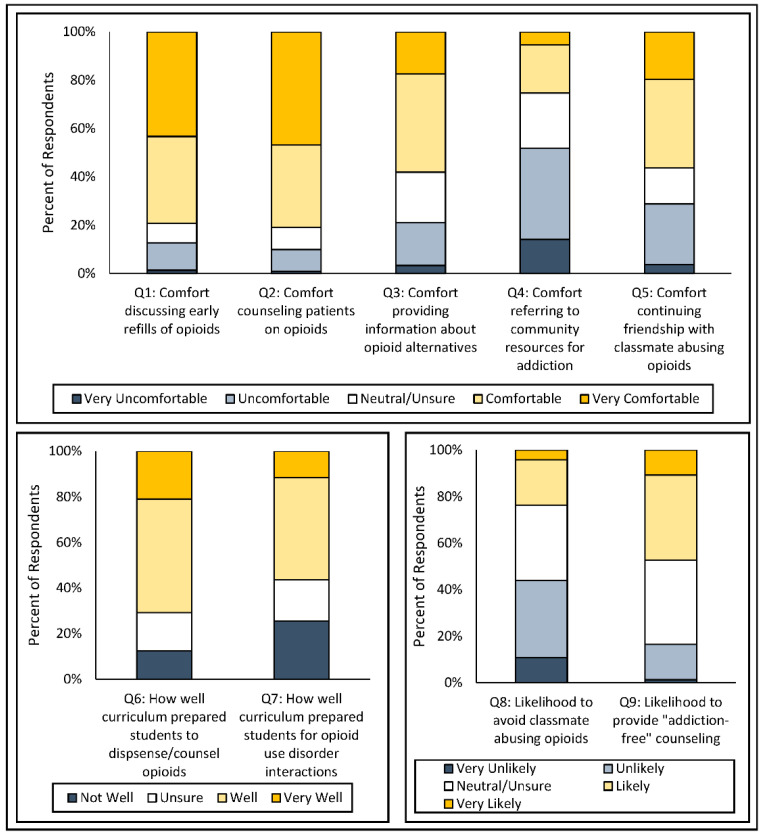
Responses from students (*n* = 244) to nine Likert-scale survey items regarding perceptions of opioid use and stigma.

**Table 1 pharmacy-08-00144-t001:** Demographics of students (*n* = 244) who responded to the survey.

Characteristic	
Age in years, median (range)	24 (20–38)
Male sex, *n* (%)	74 (30.3)
Race, *n* (%)	
Asian/Pacific Islander	32 (13.1)
Black/African American	21 (8.6)
Native American	1 (0.4)
White	177 (72.5)
Prefer not to disclose	13 (5.3)
Year in Doctor of Pharmacy Program, *n* (%)	
First (P1)	33 (13.5)
Second (P2)	82 (33.6)
Third (P3)	78 (32.0)
Fourth (P4)	51 (20.9)
Currently Employed in a Pharmacy, *n* (%)	
Yes	214 (87.7)
No	30 (13.3)
Duration of Current Employment in Months, median (range)	24 (1–108)
Pharmacy Setting of Current Employment, *n* (%)	
Community	150 (61.5)
Hospital	58 (23.8)
Other	6 (2.5)
How Often with Patients Receiving Opioid Prescriptions, *n* (%) ^a^	
Weekly	141 (94.0)
Monthly	7 (4.7)
Quarterly	1 (0.7)
Never	1 (0.7)
How Often Patients Ask if the Pharmacy Carries Certain Opioids, *n* (%) ^a^	
Weekly	87 (58.0)
Monthly	27 (18.0)
Quarterly	15 (10.0)
Never	21 (14.0)

^a^ Includes students working in community pharmacy only (*n* = 150).

**Table 2 pharmacy-08-00144-t002:** Demographics of students currently working in a community pharmacy setting versus those not working in a community pharmacy setting.

Characteristic	Community (*n* = 150)	Not in Community (*n* = 94)	*p* Value
Age, years			0.159 ^a^
Median	24	24	
Range	20–38	21–38	
Male sex, *n* (%)	39 (26.0)	35 (37.2)	0.060 ^b^
Race, *n* (%)			0.508 ^b^
Asian/Pacific Islander	20 (13.3)	12 (12.8)	
Black/African American	12 (8.0)	9 (9.6)	
Native American	1 (0.7)	0 (0)	
White	107 (71.3)	70 (74.5)	
Prefer not to disclose	10 (6.7)	3 (3.2)	
Professional Year, *n* (%)			0.100 ^b^
First	25 (16.7)	8 (8.5)	
Second	43 (28.7)	39 (41.5)	
Third	48 (32.0)	30 (31.9)	
Fourth	34 (22.7)	17 (18.1)	

^a^ Mann–Whitney; ^b^ chi-square; level of significance *p* < 0.05.

**Table 3 pharmacy-08-00144-t003:** Demographics by student’s year in Doctor of Pharmacy program.

	First Year (*n* = 33)	Second Year (*n* = 82)	Third Year (*n* = 78)	Fourth Year (*n* = 51)	*p* Value
Age, years					**< 0.001 ^a^**
Median	23	23	24	26	
Range	20–38	21–35	23–34	24–38	
Male sex, *n* (%)	7 (21.2)	25 (30.5)	20 (25.6)	22 (44.0)	0.089 ^b^
Race, *n* (%)					0.229 ^b^
Asian/Pacific Islander	3 (9.1)	13 (15.9)	10 (12.8)	6 (11.8)	
Black/African American	7 (21.2)	4 (4.9)	9 (11.5)	1 (2.0)	
Native American	0	0	1 (1.3)	0	
White	22 (66.7)	59 (72.0)	55 (70.5)	41 (80.4)	
Prefer not to disclose	1 (3.0)	6 (7.3)	3 (3.8)	3 (5.9)	
Currently working in community pharmacy setting, *n* (%)	25 (75.8)	43 (52.4)	48 (61.5)	34 (66.7)	0.100 ^b^

^a^ Kruskal–Wallis; ^b^ chi-square; level of significance *p* < 0.05.

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
