# Peer review of "Stigma of Opioid Use Disorder and Its Indirect Effects on Student Pharmacists’ Perceptions and Attitudes"

_pharmacy, 2020, doi:10.3390/pharmacy8030144_

Round 1
Reviewer 1 Report
It is a well written and interesting paper. I have some comments and suggestions.
Line 75: the name of the university should be included. How many students are enrolled at the Collage of Pharmacy and thus eligible to participate in the survey?
Line 89: should this not be likelihood instead of comfort? Since the questions are related to likelihood of interacting and the alternatives are ”very likely to very unlikely”.
Line 105: also include information regarding number of respondents.
Line 105-117: the results regarding characteristics of the responds could preferable be presented in a table in order to enhance readability.
Figure 1 needs revision. It is difficult to interpret since there are different options for the different questions. The different green colors are difficult to separate. Could perhaps questions with different options be separated into separate figures? Is it the number of students that is shown?
Line 146: how is the mean rank score obtained? Please clarify.
Line 155: continuing a friendship with a colleague. Should this not be contenting friendship with a classmate?
Table 1: what does professional year refer to? Please clarify.
How are the characteristics of the respondents in comparison with pharmacy students in general? Is the sample representative?
The results regarding the comparison between students working in retail pharmacy setting or not is not really discussed in the discussion. Please elaborate on these results in the discussion. Did the duration of employment affect the results in any way?
Another limitation could be that the students were all enrolled at the same university, thus not representative of all pharmacy students.
Could beliefs and attitudes be better explored with open-ended questions? Please also discuss the choice of method.
Line 223-224; please clarify that this is regarding opioids.
Author Response
Point 1: Line 75: the name of the university should be included. How many students are enrolled at the Collage of Pharmacy and thus eligible to participate in the survey?
Response: Thank you for this thoughtful suggestion. The text has been amended.
Point 2: Line 105: also include information regarding number of respondents.
Response: Thank you for this recommendation. The manuscript includes the information.
Point 3: Line 89: should this not be likelihood instead of comfort? Since the questions are related to likelihood of interacting and the alternatives are ”very likely to very unlikely”.
Response: Thank you for asking for this clarification. We addressed the issue.
Point 4: Line 105-117: the results regarding characteristics of the responds could preferable be presented in a table in order to enhance readability.
Response: Thank you for this recommendation. We presented the manuscript as a table, which looks much better. We really appreciate this suggestion.
Point 5: Figure 1 needs revision. It is difficult to interpret since there are different options for the different questions. The different green colors are difficult to separate. Could perhaps questions with different options be separated into separate figures? Is it the number of students that is shown?
Response: We are grateful for this recommendation. We amended the text, which is clearer.
Point 6: Line 146: how is the mean rank score obtained? Please clarify.
Response: Thank you for this clarification. We amended the text how the rank scores were obtained.
Point 7: How are the characteristics of the respondents in comparison with pharmacy students in general? Is the sample representative?
Response: We are grateful for this recommendation. The text has been amended.
Point 8: Another limitation could be that the students were all enrolled at the same university, thus not representative of all pharmacy students.
Response: We are thankful for suggesting this limitation. We amended the text to reflect it.
Point 8: Could beliefs and attitudes be better explored with open-ended questions? Please also discuss the choice of method.
Response: Thank you for this suggestion. We amended the text to reflect why the survey was selected as the main method of research.
Point 9: Line 223-224; please clarify that this is regarding opioids.
Response: We are grateful for this clarification that was addressed in the text.
Reviewer 2 Report
This manuscript describes pharmacy students’ perceptions of stigma related to opioid use disorder. This topic is important as stigma has a significant impact on the way that patients are treated, and subsequently the way that patients perceive themselves, both of which have the potential to result in negative outcomes. There is a clear justification for the importance of this study and the manuscript itself is very well-written. There are some areas of concern which should be addressed in order to clarify and strengthen the manuscript.
Methods
The survey questions are not directly about stigma and appear to be more about the indirect effects of stigma on perceptions and behaviors. What appears to be solicited is a combination of comfort with interacting with patients or friends taking opioids, predicted behaviors towards patients or friends taking opioids, and curricular effectiveness. The authors should consider the most appropriate way to represent these concepts, as the current title and way the content is presented starting in the Methods and moving through the rest of the manuscript is not entirely accurate.
Information should be provided regarding survey development, including use of items from previously-developed surveys, self-developed items (and associated face and content validity), and any pilot tests used.
Please clarify if by “addiction-free” counseling you are referring to patient counseling about opioids that does not address the potential for addiction with use?
Information should be provided about the timing of the survey. For example, if this was provided at the beginning of the academic year, P1 students would have had no professional classes and their perceptions would likely be closer to those of undergraduates.
Discussion
Information about the opioid problem in Tennessee may be better situated in the Introduction, which would allow for additional context as to why this is an even more acute educational need at UTHSC.
Some literature suggests individuals are poor predictors of their own future behavior. This limitation should be considered and discussed within the context of survey findings.
The small sample size, particularly given the fact that this took place in a single institution.
Author Response
Point 1: The survey questions are not directly about stigma and appear to be more about the indirect effects of stigma on perceptions and behaviors. What appears to be solicited is a combination of comfort with interacting with patients or friends taking opioids, predicted behaviors towards patients or friends taking opioids, and curricular effectiveness. The authors should consider the most appropriate way to represent these concepts, as the current title and way the content is presented starting in the Methods and moving through the rest of the manuscript is not entirely accurate.
Information should be provided regarding survey development, including use of items from previously-developed surveys, self-developed items (and associated face and content validity), and any pilot tests used.
Response: Thank you for these suggestions. We changed the title as you suggested, and we believe the manuscript is stronger now. This is the new title: “Stigma of opioid use disorder and its indirect effects on student pharmacists’ perceptions and attitudes.”
Point 2: Please clarify if by “addiction-free” counseling you are referring to patient counseling about opioids that does not address the potential for addiction with use?
Response: Thank you for your clarification. “Addiction-free” counseling refers to the tendency to counsel to a greater extent in those with suspected OUD.
Point 3: Information should be provided about the timing of the survey. For example, if this was provided at the beginning of the academic year, P1 students would have had no professional classes and their perceptions would likely be closer to those of undergraduates.
Response: Thank you for this suggestion. We amended the text that reflects the points made by reviewer 1.
Point 4: Information about the opioid problem in Tennessee may be better situated in the Introduction, which would allow for additional context as to why this is an even more acute educational need at UTHSC.
Response: Thank you for this recommendation. However, in the light of the 1st reviewer who did not recommend any changes to the introduction, we will keep the introduction as it is.
Point 5: Some literature suggests individuals are poor predictors of their own future behavior. This limitation should be considered and discussed within the context of survey findings.
Response: We are grateful for this suggestion. However, this survey was not intended to sample patients with opioid use disorder. Therefore, we do not think it will strengthen our discussion to debate the predictors on their own future behaviors. Indeed, if the survey would have sampled patients with OUD, then this suggestion would have been an invaluable part of the discussion.
- Could this refer to the student pharmacists who might not be able to predict their behavior when it comes to stigmatizing those with OUD in their future practice?
Point 6: The small sample size, particularly given the fact that this took place in a single institution.
Response: We are thankful for pointing out this limitation. We incorporated your recommendation and the reviewer 1 in the amended text.
Round 2
Reviewer 1 Report
Thank you for the revised manuscript. I think the authors have adressed my comments satisfactorily. Just a small comment regarding the abstract, line 26, the word pharmacy should be added at the end i.e. community pharmacy. The same goes for line 145, 147, 208 and 266, pharmacy should be included for clarification.
Author Response
Point 1: Response: Thank you for recommending to add the word pharmacy in the text. It reads much better
Reviewer 2 Report
Thanks for your thoughtful responses to reviewer comments. The enhanced description of the survey and subsequent re-organization of the Results section improved readability and provided important context for the study itself. Two remaining questions:
1) Did the question about employment only solicit if students were currently working? What about students who have had past pharmacy experience?
2) My comment on the previous review regarding individuals being poor predictors of future behavior pertained to questions 8 and 9 on the survey. Perhaps this poor ability to predict future behavior contributed to the lack of statistical difference across groups compared to other survey items?
Author Response
Point 1: Response:Past pharmacy experience was not assessed. Thank you for pointing this out! We have added some language regarding this fact to the strengths and limitations section.
Thank you for this question. The survey solicited answers from the entire class.
Point 2:Response: Thank you for clarifying. We have added a short paragraph to the discuss that addresses this.